# Provider-led community antiretroviral therapy distribution in Malawi: Retrospective cohort study of retention, viral load suppression and costs

**John Songo** [1]*, **Hannah S. Whitehead**[2], **Brooke E. Nichols** [3], **Amos Makwaya**[1], **Joseph Njala**[1], **Sam Phiri**[1,4], **Risa M. Hoffman** [2], **Kathryn Dovel**[2], **Khumbo Phiri**[1], **Joep J. van Oosterhout**[1,2]

1 Partners in Hope, Lilongwe, Malawi, 2 Division of Infectious Diseases, Department of Medicine, David Geffen School of Medicine, UCLA, Los Angeles, California, United States of America, 3 FIND, Geneva, Switzerland, 4 Department of Public Health and Family Medicine, Kamuzu University of Health Sciences, Lilongwe, Malawi

* pemphosongo@gmail.com

**Data Availability Statement:** 1. Data contain potentially identifying or sensitive participant information. Even though the data for this

## Abstract

### Background

Outcomes of community antiretroviral therapy (ART) distribution (CAD), in which provider–led ART teams deliver integrated HIV services at health posts in communities, have been mixed in sub-Saharan African countries. CAD outcomes and costs relative to facility-based care have not been reported from Malawi.

### Methods

We performed a retrospective cohort study in two Malawian districts (Lilongwe and Chikwawa districts), comparing CAD with facility-based ART care. We selected an equal number of clients in CAD and facility-based care who were aged >13 years, had an undetectable viral load (VL) result in the last year and were stable on first-line ART for ≥1 year. We compared retention in care (alive and no period of ≥60 days without ART) using Kaplan-Meier survival analysis and Cox regression and maintenance of VL suppression (<1,000 copies/mL) during follow-up using logistic regression. We also compared costs (in US$) from the health system and client perspectives for the two models of care. Data were collected in October and November 2020.

### Results

700 ART clients (350 CAD, 350 facility-based) were included. The median age was 43 years (IQR 36–51), median duration on ART was 7 years (IQR 4–9), and 75% were female. Retention in care did not differ significantly between clients in CAD (89.4% retained) and facility-based care (89.3%), p = 0.95. No significant difference in maintenance of VL suppression were observed between CAD and facility-based care (aOR: 1.24, 95% CI: 0.47–3.20, p = 0.70). CAD resulted in slightly higher health system costs than facility-based care: $118/

manuscript are de-identified, the information that the study was conducted at Partners in Hope's supported sites may draw attention to individual clients attending ART clinics and health care workers at the health facilities and associated CAD spokes. 2. Who has imposed the restriction of publicly sharing the manuscript data? The Malawi's National Health Sciences Research Committee approved our protocol in which under the ethical considerations section on protection of human subjects' privacy and confidentiality, we stated that participant information will not be released without written permission from the researchers, except as necessary for review, monitoring, and/or auditing. 3. Provide non-author contact information for a data access committee, ethics committee, or other institutional body to which data requests may be sent. Our non-author institutional representative who would be able to hold the data and respond to external requests for data access is the PIH Director of Monitoring and Evaluation and their contact information is provided below. Mackenzie Chivwara, mackenzie@pihmalawi.com Partners In Hope, Area 36 Plot 8, Kamuzu Procession Road Opposite, Lilongwe.

**Funding:** This study was supported by the United States Agency for International Development [Cooperative Agreement AID-OAA-A-15-00070]. Cooperative Agreement No: 72061221CA00010. The views in this publication do not necessarily reflect the views of the U. S. Agency for International Development (USAID), the U. S. President's Emergency Plan for AIDS Relief (PEPFAR) or the United States Government. The funders had no role in study design, data collection and analysis, decision to publish, or preparation of the manuscript.

**Competing interests:** The authors have declared that no competing interests exist.

year vs. $108/year per person accessing care; and $133/year vs. $122/year per person retained in care. CAD decreased individual client costs compared to facility-based care: $3.20/year vs. $11.40/year per person accessing care; and $3.60/year vs. $12.90/year per person retained in care.

## Conclusion

Clients in provider-led CAD care in Malawi had very good retention in care and VL suppression outcomes, similar to clients receiving facility-based care. While health system costs were somewhat higher with CAD, costs for clients were reduced substantially. More research is needed to understand the impact of other differentiated service delivery models on costs for the health system and clients.

## Introduction

Antiretroviral therapy (ART) coverage has expanded greatly in sub-Saharan Africa and Malawi over the past 20 years [1]. The sharply risen number of ART clients has resulted in health system challenges such as long waiting times, insufficient staffing and limited space [2]. Together with the need for more client-centered and tailored care, such challenges have inspired innovative ways of delivering care to clients, broadly referred to as differentiated service delivery (DSD) models [3]. Provider-led and community-led Community ART Distribution (CAD) DSD models have been implemented in Sub-Saharan Africa since 2010 [4]. In provider-led CAD, a team of health care workers (HCWs) travels to small, community-based health posts to provide comprehensive HIV services, including dispensing ART refills, on scheduled days. In community-led CAD, ART clients form groups and one community member collects ART refills at the facility for the whole group [5]. Compared to facility-based services, CAD brings HIV care closer to people's homes, thereby reducing travel time, travel costs and opportunity costs, which may improve adherence and facilitate retention in care [3, 6].

Despite widespread implementation of CAD across sub-Saharan Africa, evidence regarding its effectiveness and cost-effectiveness in improving retention and viral load (VL) suppression is limited, outcomes have varied and none have been reported from Malawi. An early systematic review of studies from sub-Saharan Africa suggested that retention in care is higher in provider- and community-led CAD than in standard facility-based care [6], but two later randomized trials in Zimbabwe and Lesotho found that retention in care and VL suppression did not differ significantly between health facility-based care, community-led CAD, and provider-led CAD [7, 8]. Compared to standard facility-based care, CAD models had higher health system cost per client retained in care at 12 months in Zambia [9].

Comparing retrospective cohort data from clients receiving provider-led CAD and clients receiving facility-based care, we sought to evaluate the impact of provider-led CAD services on retention and VL suppression in Malawi. We also assess differences in cost between provider-led CAD services and facility-based care from a health systems and individual client perspective.

## Methods

### Study setting

Partners In Hope (PIH) is a Malawian non-governmental organization that provides PEPFAR-funded support for HIV services in Malawi's National HIV program. PIH implements a

provider-led CAD model for stable (suppressed VL <1,000 copies/mL and on first-line ART) individuals at 4 health facilities in Lilongwe (urban setting, n = 2) and Chikwawa (rural setting, n = 2) districts, with a total of 20 CAD outreach sites. Recent population-based estimates indicate that HIV prevalence is 9.0% in Lilongwe city and 13.3% in southwestern Malawi (where Chikwawa is located) [10].

Provider-led CAD follows a hub-and-spoke model where a team of HCWs travels once a month from the hub (a large health facility) to a spoke (a village-based health post). The team is generally led by a nurse or occasionally a clinical officer, and includes an HIV diagnostic assistant (HDA), Treatment Supporter, counsellor and driver. Once a quarter, a data clerk joins the team. Services provided at CAD spokes include ART refills, VL sample collection, HIV testing services, family planning, screening for non-communicable diseases, referral to community services and adherence assessment and counselling.

## Study design and data collection

We conducted a retrospective cohort study making use of routinely collected data from standard medical records of 700 ART clients receiving ART care through CAD or health facility-based services. Data were collected in October-November 2020, from two 'hub' health facilities in Chikwawa and two in Lilongwe, and 20 associated CAD 'spoke' sites. We selected all the clients registered in CAD between January 2019 (when CAD implementation began) and June 2019, and collected data from their first visit through August 2020 to allow for 14 months of observation time. Clients can choose to be enrolled into provider-led CAD if they meet programmatic criteria: being stable on first-line ART for more than 12 months, registered at a hub site, ready to disclose HIV status to other members of the CAD spoke, undetectable result of the last VL test (within 12 months), and age 13 years or older. We then compiled a list of individuals receiving standard care at the four hub facilities ("controls"). The inclusion criteria for controls were the same as for CAD enrolment mentioned above, except for the disclosure criterion. The number of controls included for each hub was equal to the number of CAD clients enrolled from that hub facility's CAD spokes. We selected hub controls chronologically, starting with those who had visited the facility in January 2019, and proceeding until the required number of clients was reached.

We extracted data on all recorded visits for 14 months after study entry from clients' individual ART master-cards. A 14-month period was used to be able to capture full 1-year outcomes, including defaulting, as defined below. Collected data included: demographic characteristics, ART initiation date, dates of all CAD/facility visits, results of annual routine VL tests, and standardized ART outcomes (default, transfer out, stopped ART, death). All data were collected on Android tablets using the SurveyCTO electronic data collection platform. Additional VL test results were extracted from the National Laboratory Information Systems Management (LIMS) Database, if missing on ART master-cards. No personal identifiers were collected from the data sources and therefore all data were anonymized. Ethical approval for the study was obtained from the National Health Sciences Research Committee in Malawi (#1099).

## Key study definitions

**Adverse outcome.** Recorded to have died, defaulted or stopped ART during follow-up. For individuals with multiple adverse outcomes recorded, the earliest outcome was used for all analyses. For example, if a client had defaulted, then returned to care, and subsequently died, they were classified as having defaulted. Clients who did not experience the adverse outcome were classified as *retained* after censoring for transfer outs.

**Defaulted.**   Per national guidelines, defaulting from care was defined as being overdue for an ART refill appointment and estimated to have run out of ARVs for 2 months or longer, based on the quantity of ARVs dispensed at the last visit [11].

**Viral load outcomes.**   VL results were categorized according to national HIV guidelines: high VL ($\geq$1000 copies/ml); low level viremia (200–999 copies/mL); or suppressed VL (undetectable-199 copies/mL). For logistic regression analyses, we applied a commonly-used binary VL outcome: <1000 copies/mL and $\geq$1000 copies/mL [12]. VL results were only included from samples taken at least 6 months and no more than 18 months after enrollment into CAD or from the first visit in the study period for controls. We extended the period for VL results from 14 to 18 months due to the infrequency of VL testing during the study period.

## Statistical analysis

Descriptive statistics were generated for client characteristics and outcomes. We compared viral load outcomes between the two models of care using chi-square tests and logistic regression. We used Kaplan-Meier survival analysis methods to compare retention in care at CAD spokes versus hub facilities over the 14 months of follow-up. Each client started contributing person-time to the analysis from the day that they were in enrolled into CAD or the earliest day that they visited the hub facility between January 2019 and June 2019. Follow-up time of clients who transferred to another health facility and of those who remained in care at the end of the follow-up period was censored on their last recorded visit day within the 14 months follow up period. We used Cox regression to produce unadjusted and adjusted hazard ratios (aHR) of experiencing an adverse outcome, adjusting for sex, age and district.

Univariate analysis of the association between key outcome variables and the intervention were stratified by gender to take into account significant gender differences across the HIV care cascade in Malawi [13].

We assessed the extent of missing data in the sample by key outcome variables and covariates (retention, VL suppression, sex, age and district). Only VL suppression had missing data (21%) but the missing data did not vary by arm of intervention (CAD/hub) (p = 0.09). All observations with missing data on VL suppression were excluded from the analysis on VL suppression.

All analyses were conducted with Stata v17 (Stata Corp., Texas, USA).

## Cost analysis

We calculated the cost per person retained for both facility-based care and CAD care from a health systems and individual client perspective.

Table 1 describes the unit costs collected locally or derived from the literature for both the cost to the health system and the cost to the client. For health systems cost per CAD spoke clinic day, we include all additional activities required to successfully implement a CAD visit, including use of a vehicle to and from the CAD (average of 32km round-trip per CAD visit), and the cost of a day of the full clinical team's services. The total costs were divided by the number of visiting clients to each CAD included in the study to determine the cost per individual client CAD visit.

## Results

### Sample characteristics at enrolment

We collected data on 700 ART clients, 350 from provider-led CAD and 350 from facility care (Table 2). Approximately 75% of both CAD and hub clients were female. Clients at CAD spokes were older (45.3 vs. 42.2 years) and had been on ART for longer (7.2 vs. 6.4 years).

**Table 1. Unit costs included in analysis.**

| Variable | Unit cost (USD) 2022 | Sources |
|---|---|---|
| Provider costs | | |
| Hub clinic visit | $2.90 | [14] |
| CAD spoke visit | $5.95 | Program implementation data (2022) |
| ART day (first-line) | $0.21 | [14] |
| Viral load | $14.2 | [14] |
| Cost to client | | |
| Wage lost for hub facility visit* | $2.50 | In-country information |
| Wages lost for CAD spoke visit** | $0.625 | In-country information |

*1 day of minimum wage lost;

**0.25 day of minimum wage lost

## Client outcomes

**Descriptive outcomes at 14 months of follow up.** At the end of the 14-months of follow-up, approximately 1% of participants had died, 10% had defaulted, and 5% had transferred to another health facility. Differences in standard client outcomes between CAD and facility-based care were minimal and not statistically significant, either overall or when stratified by sex (Table 3).

**Kaplan-Meier survival estimates of retention in care at 14 months of follow up.** The average follow-up time was 11.6 months for clients at CAD spokes and 11.1 months at hub facilities. The cumulative probability of retention in care over the follow-up period was not

**Table 2. Characteristics of clients at enrollment in CAD spokes and at hub health facilities (controls).**

| Variable | Overall | | CAD spokes | | Hub facilities | |
|---|---|---|---|---|---|---|
| | n | % | n | % | n | % |
| Total | 700 | 100 | 350 | 50.0 | 350 | 50.0 |
| Age | | | | | | |
| Median, years (IQR) | 43.8 (36.5–51.4) | | 45.3 (38.3–54.1) | | 42.2 (34.6–49.4) | |
| 13–24 years | 27 | 3.9 | 10 | 2.8 | 17 | 4.9 |
| 25–34 years | 111 | 15.9 | 38 | 10.8 | 73 | 20.9 |
| 35–44 years | 246 | 35.1 | 123 | 35.1 | 123 | 35.1 |
| 45–54 years | 182 | 26.0 | 97 | 27.7 | 85 | 24.2 |
| 55+ years | 134 | 19.1 | 82 | 23.4 | 52 | 14.9 |
| Sex | | | | | | |
| Male | 178 | 25.4 | 87 | 24.9 | 91 | 26.0 |
| Female | 522 | 74.6 | 263 | 75.1 | 259 | 74.0 |
| Duration on ART | | | | | | |
| Median (IQR) | 6.8 (4.1–9.7) | | 7.2 (4.2–9.9) | | 6.4 (3.9–9.3) | |
| <4year | 171 | 24.4 | 81 | 23.1 | 90 | 25.7 |
| 4-8years | 251 | 35.9 | 119 | 34.0 | 132 | 37.7 |
| >8years | 278 | 39.7 | 150 | 42.9 | 128 | 36.6 |
| District | | | | | | |
| Lilongwe | 228 | 32.6 | 114 | 32.6 | 114 | 32.6 |
| Chikwawa | 472 | 67.4 | 236 | 67.4 | 236 | 67.4 |

**Table 3. Client outcomes at 14 months.**

| Outcome | Overall | | | | | Males | | | | | Females | | | | |
|---|---|---|---|---|---|---|---|---|---|---|---|---|---|---|---|
| | CAD | | Hub | | p-value | CAD | | Hub | | p-value | CAD | | Hub | | p-value |
| | n | % | n | % | | n | % | n | % | | n | % | n | % | |
| Died | 4 | 1.1 | 3 | 0.9 | 0.619 | 3 | 3.4 | 2 | 2.2 | 0.832 | 1 | 0.4 | 1 | 0.4 | 0.387 |
| Stopped ART | 0 | 0.0 | 0 | 0.0 | | 0 | 0.0 | 0 | 0.0 | | 0 | 0.0 | 0 | 0.0 | |
| Defaulted | 35 | 10.0 | 34 | 9.7 | | 13 | 14.9 | 11 | 12.1 | | 22 | 8.4 | 23 | 8.9 | |
| Transferred Out | 12 | 3.4 | 19 | 5.4 | | 4 | 4.6 | 3 | 3.3 | | 8 | 3.0 | 16 | 6.2 | |
| Alive on ART | 299 | 85.4 | 294 | 84.0 | | 67 | 77.0 | 75 | 82.4 | | 232 | 88.2 | 219 | 84.6 | |
| Total | 350 | 100 | 350 | 100 | | 87 | 100 | 91 | 100 | | 263 | 100 | 259 | 100 | |

significantly different between clients in provider-led CAD and facility-based HIV care (88.4% vs. 88.3%; p-value 0.95) (Fig 1). Sex was the only covariate with a statistically significant impact on retention in the total study population (males 80.4% vs. females 90.8%; p = 0.005) (Fig 2).

Controlling for sex, age, and district of residence, the risk of experiencing an adverse outcome was similar for clients in provider-led CAD and facility-based care (aHR: 1.05, 95%CI: 0.66–1.66, P-value: 0.80)

## Viral load outcomes

VL test results were available for 553 clients, representing 79% of the overall sample (82% of CAD clients; 76% of hub clients; p = 0.09). In univariable analyses, standard VL outcomes

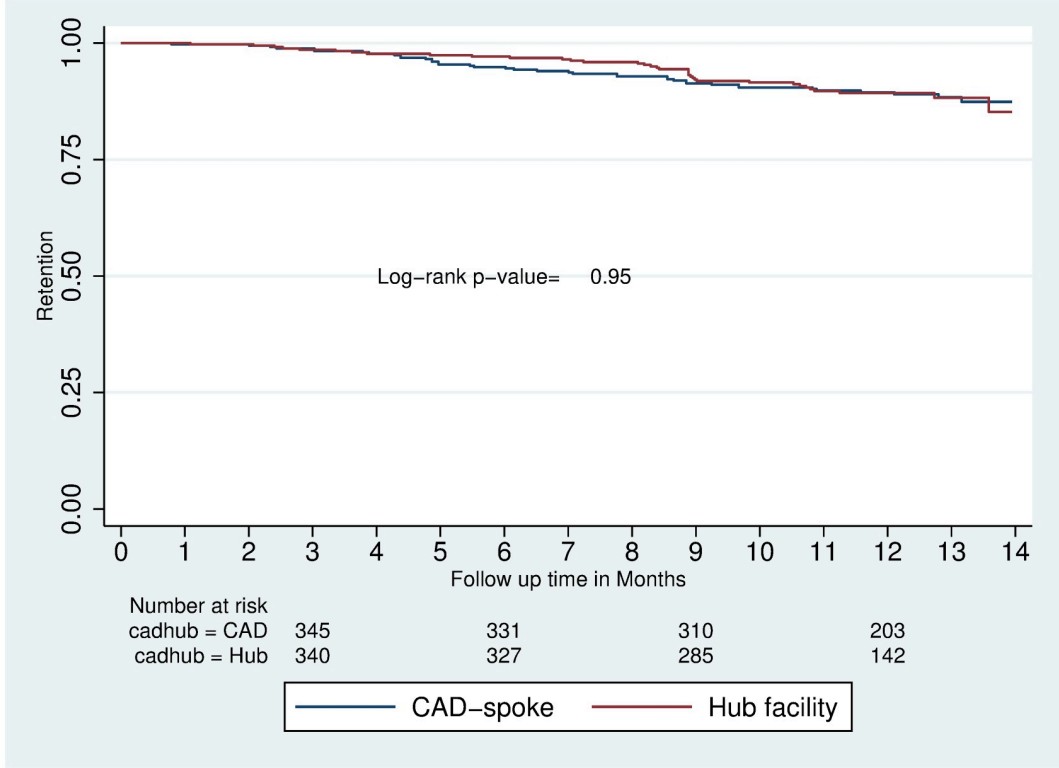

**Fig 1. Kaplan-Meier survival estimates of retention over 14 months of follow up by model of care.**

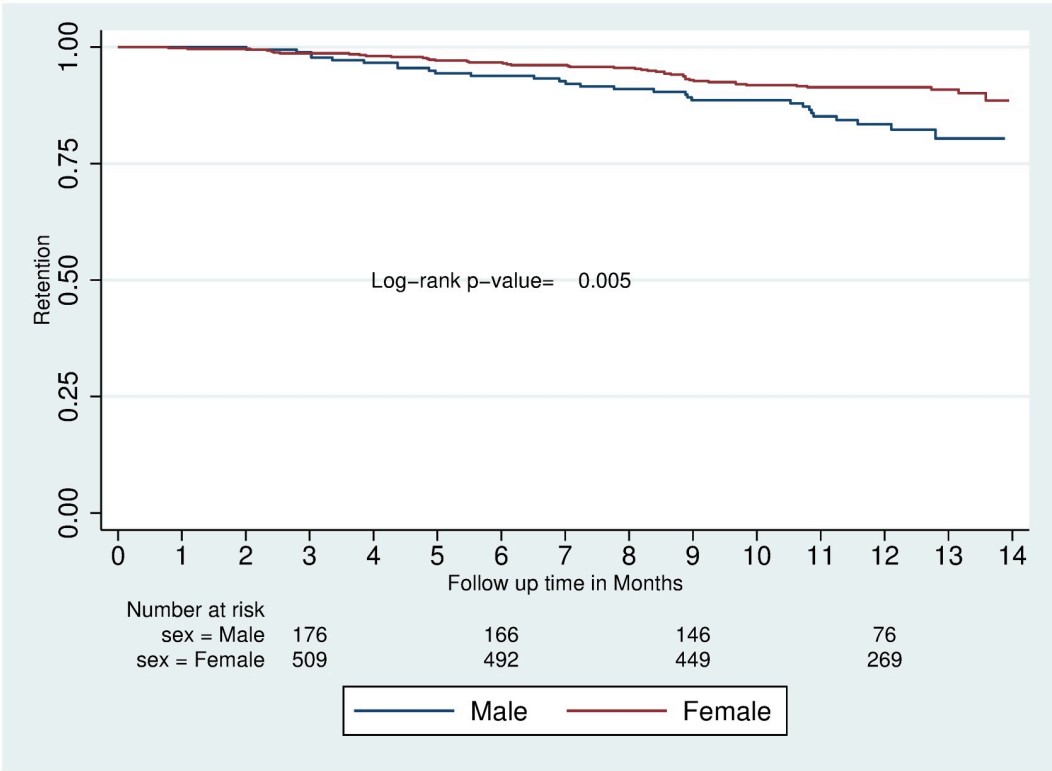

**Fig 2. Kaplan-Meier survival estimates of retention over 14 months of follow up by gender.**

were not significantly different between CAD and health facility-based care, whether overall or stratified by sex (Table 4). After adjusting for sex, duration on ART, age, and district of residence, there was no significant difference in prevalence of VL <1,000 copies/ml between clients in CAD (97%) and in hub care (96%): aOR 1.24, 95% CI 0.47–3.26, p-value 0.66.

## Costs

Table 5 describes ART services utilization per client for both CAD-based and facility-based care. Differences in resource use, considering the standard deviations, were small between the models of care.

The total costs per person receiving care and the costs per person retained by model of care, from a health systems and individual client perspective, are described in Table 6. Health system cost estimates for provider-led CAD were $118 per person receiving care per year and

**Table 4. Viral load suppression.**

| VL | Overall | | | | | Males | | | | | Females | | | | |
|---|---|---|---|---|---|---|---|---|---|---|---|---|---|---|---|
| | CAD | | Hub | | p-value | CAD | | Hub | | p-value | CAD | | Hub | | p-value |
| | n | % | n | % | | n | % | n | % | | n | % | n | % | |
| ≥1000 copies/ml | 8 | 2.8 | 10 | 3.7 | 0.260 | 2 | 3.0 | 2 | 3.1 | 0.264 | 6 | 2.7 | 8 | 3.9 | 0.552 |
| 200–999 copies/mL | 57 | 20.0 | 40 | 14.9 | | 14 | 21.2 | 7 | 10.8 | | 43 | 19.6 | 33 | 16.3 | |
| undetectable-199 copies/mL | 220 | 77.2 | 218 | 81.3 | | 50 | 75.8 | 56 | 86.2 | | 170 | 77.6 | 162 | 79.8 | |
| Total | 285 | 100 | 268 | 100 | | 66 | 100 | 65 | 100 | | 219 | 100 | 203 | 100 | |

**Table 5. ART services utilization of CAD-based care and facility-based care in Malawi (average and standard deviation).**

| Variable | CAD-based care | Facility-based care |
|---|---|---|
| Number of clinic visits | 0 | 4.6 (SD 1.4) |
| Number of CAD visits | 5.1 (SD 1.5) | 0 |
| Number of ART days | 351 (SD 92) | 383 (SD 91) |
| Number of viral load tests | 0.9 (SD 0.7) | 1.0 (SD 0.6) |

**Table 6. Cost per person in care and cost per person retained comparing facility- and provider-led CAD- (based on 2022 USD).**

| Variable | CAD | Facility-based care |
|---|---|---|
| Health system cost | | |
| Cost per client | $118 (SD $29) | $108 (SD $24) |
| Cost per client retained | 133 | 122 |
| Client cost | | |
| Cost per client | $3.17 (SD $0.96) | $11.44 (SD $3.53) |
| Cost per client retained | 3.58 | 12.88 |

$108 per person per year in facility-based care. Cost for individual clients was lower in provider-led CAD ($3.17 per person per year) than facility-based care ($11.44 per person per year). In a second analysis limited to those retained in care, the gap in health systems costs between the two models of care was the same ($10 lower in provider-led CAD), while differences in costs for clients widened slightly (from $8.27 reduced cost per client to $9.30 reduced cost for client retained in provider-led CAD).

## Discussion

In a retrospective assessment of two models of HIV care in Malawi, we found that retention in care and viral suppression outcomes were similar in provider-led CAD as compared to facility-based care. While health system cost per person provided care and per person retained in care were 9% higher in provider-led CAD, cost for 12-months of HIV care incurred by clients was 72% lower in CAD than in facility-based care.

Prior studies of CAD in sub-Saharan Africa had mixed findings. In two similar cluster randomized trials, one in Lesotho and one in Zimbabwe, no significant differences in retention in care and VL suppression were observed between facility-based 3-monthly ART (control), 3-monthly community ART groups, and 6-monthly provided-led CAD [7, 8]. In contrast, in a retrospective cohort study from Mozambique, retention in care was significantly higher among clients in community-led CAD (at 99.1% at 12 months) than in facility-based care (89.5%, p<0.0001) [15]. In the same setting, another retrospective cohort study found that individuals in facility-based care were more than twice as likely to be lost to follow up (HR 2.356; p = 0.04) than matched participants in community-led CAD [5]. Differences in the organization of CAD services, variations in the local setting (including rural vs. urban) and the degree of donor support may explain these variations in ART outcomes. It has been suggested that in community-led CADs, clients acquire ownership of their ART care, as group representatives take turns to collect ART at the facility for the whole group and members actively trace group members who miss a group ART distribution session [15]. Such activities may increase group members' ability to self-manage their

own care [16] and lead to better retention and VL suppression. To date, CADs are largely only available to stable ART clients and this presents potential missed opportunities to engage non-stable clients in CAD.

Our findings showed that men had lower retention in comparison to women, in line with earlier findings that men have poorer outcomes across the HIV care cascade in sub-Saharan Africa [13, 17]. Social norms of masculine physical strength, self-reliance, and beliefs that clinics are women's spaces are some reasons for men's lower retention in HIV services [18, 19].

From a health systems perspective, provider-led CAD is slightly more expensive than facility-based care. The difference was relatively small (9% higher, or approximately $10 more per person receiving care per year). Additional costs are due to HCWs traveling to CAD locations, but the vast majority of ART costs are generated by medication and VL testing, which do not vary by provider-led CAD or facility-based care. However, we found that provider-led CAD substantially reduced costs for clients by nearly three-quarters, mainly due to lower opportunity costs as clients need less travel time to access ART services. Health system costs of provider-led CAD can be reduced by decreasing the frequency of the team's travels to distribution sites, in combination with expansion of multi-month dispensing. This may also further reduce clients' costs and could benefit retention and viral suppression outcomes [20]. Our results are consistent with findings from the only other study from the region that assessed costs and outcomes across community-based ART models. In this study from Zambia, costs for CAD models ranged from an annual $116/person to $199/person compared with $100/person for facility-care [9]. Considering the difference in costs for the health system between CAD and facility-based care, we estimate that scaling up CAD services by 20% in Malawi would increase the budget of HIV services for 935,000 individuals on ART in Malawi at the end of 2022 by 1,870,000 USD per year (935,000x 20% x 10 USD).

A limitation of our cost analysis is that we did not incorporate travel costs for clients (cost of transportation to/from facility-based care) as many clients walk, use their own bicycle or make informal arrangements that are not directly tied to financial cost. It is therefore highly likely that we underestimated the difference in clients' costs and provider-led CAD may have a greater financial benefit to clients than reported here. This is supported by a qualitative study on client and nurse perspectives of provider-led community-based models of HIV care in Malawi, where clients reported that CAD services resulted in savings on transportation costs and the time it took them to travel to a health facility [21].

As is common with observational studies, our results may be prone to bias as clients enrolled in CAD are selected based on specific characteristics, such as being clinically stable, which cannot be completely adjusted for in statistical analyses. However, great effort was put into ensuring that the eligibility criteria used for enrolment into CAD is adhered to during selection of controls through a strict study enrolment protocol that was double checked at data collection and at analysis.

## Conclusions

Provider-led CAD services in Malawi had excellent one-year retention and VL suppression results that were similar to facility-based care. CAD was associated with a small increase in the financial costs to the health system but substantial savings for clients, which may benefit longer term engagement in care and ART outcomes. More study is needed to determine cost-effectiveness of different DSD models in sub-Saharan Africa.

## Supporting information

**S1 Table. Unadjusted and adjusted hazard ratios (HR) of experiencing an adverse outcome over follow-up period.**
(DOCX)

**S2 Table. Unadjusted and adjusted odds ratio (OR) of maintaining viral load suppression at end of follow-up.**
(DOCX)

**S1 Checklist. STROBE checklist.**
(DOCX)

**S1 File. Ethics approval letter.**
(PDF)

## Acknowledgments

We thank the support of the participating health facilities and their management teams.

## Author Contributions

**Conceptualization:** John Songo, Sam Phiri, Risa M. Hoffman, Kathryn Dovel, Khumbo Phiri, Joep J. van Oosterhout.

**Data curation:** John Songo, Hannah S. Whitehead, Brooke E. Nichols.

**Formal analysis:** John Songo.

**Funding acquisition:** Kathryn Dovel.

**Investigation:** John Songo.

**Methodology:** John Songo, Hannah S. Whitehead, Brooke E. Nichols, Kathryn Dovel, Khumbo Phiri, Joep J. van Oosterhout.

**Supervision:** Joep J. van Oosterhout.

**Writing – original draft:** John Songo.

**Writing – review & editing:** Hannah S. Whitehead, Amos Makwaya, Joseph Njala, Sam Phiri, Risa M. Hoffman, Kathryn Dovel, Khumbo Phiri, Joep J. van Oosterhout.

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
