## [Decision Letter · Decision Letter 0]

10 Jul 2023

PGPH-D-23-00966

Provider-led community Antiretroviral Therapy distribution in Malawi: Retrospective cohort study of retention, viral load suppression, and costs.

Dear Dr. Songo,

Thank you for submitting your manuscript to PLOS Global Public Health. After careful consideration, we feel that it has merit but does not fully meet PLOS Global Public Health’s publication criteria as it currently stands. Therefore, we invite you to submit a revised version of the manuscript that addresses the points raised during the review process.

We look forward to receiving your revised manuscript.

Kind regards,

Javier H Eslava-Schmalbach, M.D., Ph.D., MSc

Academic Editor

Journal Requirements:

2. In the Funding Information you indicated that no funding was received. Please revise the Funding Information field to reflect funding received.

3. In the online submission form, you indicated that "The data for this manuscript will be made available to the to the editorial team if need

be upon request through the corresponding author". All PLOS journals now require all data underlying the findings described in their manuscript to be freely available to other researchers, either 1. In a public repository, 2. Within the manuscript itself, or 3. Uploaded as supplementary information.

Additional Editor Comments (if provided):

Dear authors

We have received the reviewers' comments. Please comment on and address each of their suggestions, incorporating them into the text.

Reviewers' comments:

Reviewer's Responses to Questions

**Comments to the Author**

1. Does this manuscript meet PLOS Global Public Health’s publication criteria? Is the manuscript technically sound, and do the data support the conclusions? The manuscript must describe methodologically and ethically rigorous research with conclusions that are appropriately drawn based on the data presented.

Reviewer #1: Yes

Reviewer #2: Yes

Reviewer #3: Yes

2. Has the statistical analysis been performed appropriately and rigorously?

Reviewer #1: Yes

Reviewer #2: Yes

Reviewer #3: Yes

3. Have the authors made all data underlying the findings in their manuscript fully available (please refer to the Data Availability Statement at the start of the manuscript PDF file)?

Reviewer #1: Yes

Reviewer #2: Yes

Reviewer #3: No

4. Is the manuscript presented in an intelligible fashion and written in standard English?

Reviewer #1: Yes

Reviewer #2: Yes

Reviewer #3: Yes

5. Review Comments to the Author

Reviewer #1: This is a well-conducted study and addresses an important research question. The manuscript is well-written. I have a few questions for the authros:

1. Was propensity matching considered for the analysis of outcomes?

2. Men were more likely to default than women. Could the authors address this finding in the discussion?

3. The authors have looked at costs at the individual level. Is it possible to comment on the budgetary / policy implications of these additional costs?

4. On what basis was the sample of 700 individuals chosen?

Reviewer #2: This is an excellent piece of work. It holds significant importance in the field of HIV/AIDS management and healthcare delivery in resource-limited settings. This study examines the impact of a community-based approach to antiretroviral therapy (ART) distribution in Malawi and evaluates its effectiveness in terms of retention in care, viral load suppression, and cost implications.I would like to congratulate the authors on this effort. The manuscript is organized and well-written. I have only two minor comments

Study setting: Add a few lines on disease burden in the two districts of Malawi.

Table 5: Please define what is meant by ‘resources’ in line 200

Reviewer #3: Thank you for the opportunity to review “Provider-led community Antiretroviral Therapy distribution in Malawi: Retrospective cohort study of retention, viral load suppression, and costs.” This is a well-written manuscript with important implications.

The following are a few suggestions for the authors to consider.

1) The definition of retention is missing from the methods. Based on the KM curves, it looks like clients were only at risk of becoming lost if they remained in care for at least 2 months. Please confirm this definition. It seems odd that no one died or was transferred within the first 2 months.

2) A table with the univariate and multivariate logistic regression models would be beneficial to include.

3) Line 136, change “crude” to unadjusted

4) Table 5 Resource utilization of CAD-based care and facility-based care in Malawi instead of reporting average and standard it may be more informative to report median (IQR)

5) Authors should elaborate on the limitations of their study. There is a possibility of a selection bias as well based on the eligibility criteria of the program. Authors can also try imputing the missing HIV viral values instead of assuming they are missing completely at random (mice command in STATA).

6) Figure legends are missing.

6. PLOS authors have the option to publish the peer review history of their article (what does this mean?). If published, this will include your full peer review and any attached files.

**Do you want your identity to be public for this peer review?** For information about this choice, including consent withdrawal, please see our Privacy Policy.

Reviewer #1: No

Reviewer #2: **Yes: **Shifa Salman Habib

Reviewer #3: No

---

## [Editor Report · Decision Letter 1]

24 Aug 2023

Provider-led community Antiretroviral Therapy distribution in Malawi: Retrospective cohort study of retention, viral load suppression, and costs.

PGPH-D-23-00966R1

Dear Mr Songo,

We are pleased to inform you that your manuscript 'Provider-led community Antiretroviral Therapy distribution in Malawi: Retrospective cohort study of retention, viral load suppression, and costs.' has been provisionally accepted for publication in PLOS Global Public Health.

Best regards,

Javier H Eslava-Schmalbach, M.D., Ph.D., MSc

Academic Editor